

# PEST sequences from a cactus dehydrin regulate its proteolytic degradation

Adriana L. Salazar-Retana[1,*], Israel Maruri-López[1,3,*], Itzell E. Hernández-Sánchez[1,4], Alicia Becerra-Flora[1], María de la Luz Guerrero-González[2] and Juan Francisco Jiménez-Bremont[1]

[1] Laboratorio de Biotecnología Molecular de Plantas, División de Biología Molecular, Instituto Potosino de Investigación Científica y Tecnológica AC, San Luis Potosí, San Luis Potosí, México
[2] Laboratorio de Biotecnología, Facultad de Agronomía y Veterinaria, Universidad Autónoma de San Luis Potosí, San Luis Potosí, México
[3] Current affiliation: Centro de Ciencias Genomicas, Universidad Nacional Autonoma de Mexico, Cuernavaca, Morelos, Mexico
[4] Current affiliation: Max-Planck-Institute of Molecular Plant Physiology, Am Mühlenberg, Potsdam, Germany
[*] These authors contributed equally to this work.

Corresponding authors
Israel Maruri-López,
ismaruri@ccg.unam.mx
Juan Francisco Jiménez-Bremont,
jbremont@ipicyt.edu.mx

## ABSTRACT

Dehydrins (DHNs) are intrinsically disordered proteins expressed under cellular dehydration-related stresses. In this study, we identified potential proteolytic PEST sequences located at the central and C-terminal regions from the *Opuntia streptacantha* OpsDHN1 protein. In order to evaluate these PEST sequences as proteolytic tags, we generated a translational fusion with the GUS reporter protein and OpsDHN1 coding sequence. We found a GUS degradation effect in tobacco agro-infiltrated leaves and Arabidopsis transgenic lines that expressed the fusion GUS::OpsDHN1 full-length. Also, two additional translational fusions between OpsDHN1 protein fragments that include the central (GUS::PEST-1) or the C-terminal (GUS::PEST-2) PEST sequences were able to decrease the GUS activity, with PEST-2 showing the greatest reduction in GUS activity. GUS signal was abated when the OpsDHN1 fragment that includes both PEST sequences (GUS::PEST-1-2) were fused to GUS. Treatment with the MG132 proteasome inhibitor attenuated the PEST-mediated GUS degradation. Point mutations of phosphorylatable residues in PEST sequences reestablished GUS signal, hence these sequences are important during protein degradation. Finally, *in silico* analysis identified potential PEST sequences in other plant DHNs. This is the first study reporting presence of PEST motifs in dehydrins.

## INTRODUCTION

The accumulation of late embryogenesis abundant (LEA) proteins has been reported as a common mechanism developed to face stress in various organisms, including algae, bacteria, yeast, and plants (*Battaglia et al., 2008*). Particularly, dehydrins (DHNs), also known as group 2 LEA proteins, are involved in the plant response and adaptation mechanisms against abiotic stresses, such as low temperatures, high salinity, and drought (*Ochoa-Alfaro et al., 2012*; *Ruibal et al., 2012*; *Muñoz Mayor et al., 2012*).

Three conserved motifs have been recognized in DHN sequences: the Y-, S-, and K-segments. The canonical characteristic among DHNs is the ubiquity of at least one K-segment (EKKGIMDKIKEKLPG), which has been related to their protective functions (*Koag et al., 2009*). The S-segment (LHRSGSSSSSSSEDD) and Y-segment (T/VDEYGNP) are also present in DHNs sequences, but do not necessarily occur in all DHNs. However, the functions of these segments are not completely understood. Recently, an 11-residue amino acid sequence (DRGLFDFLGKK) named the F-segment has been reported in the $SK_n$ class, reclassifying it as $FSK_n$ type; structural modeling suggests that this new F-segment could form amphipathic helices that may be implicated in membrane or protein binding, analogous to the K-segment (*Richard Strimbeck, 2017*). Besides these common segments, the existence of histidine-rich motifs (H-X3-H, HH, and $H_n$) has been seen in DHN sequences, and it has been proposed that histidine participates in DNA binding and as an ion chelator (*Hara, Fujinaga & Kuboi, 2005*; *Hara, Kondo & Kato, 2013*). According to their protein architecture, DHNs are grouped into five classes: $Y_nSK_n$, $K_n$, $K_nS$, $SK_n$, and $Y_nK_n$ (*Danyluk et al., 1994*); and based on the amino acid composition of DHNs, these proteins can also be classified as acidic or basic (*Mouillon, Gustafsson & Harryson, 2006*).

Another feature of DHNs is the amino acid composition, in which these proteins contain an abundance of amino acids (proline, glutamic acid, lysine, serine, and glutamine), and have a low content of hydrophobic residues (*Graether & Boddington, 2014*). This amino acid composition is analogous to those found in intrinsically disordered proteins (IDPs) (*Uversky, 2002*). Compared to ordered proteins, IDPs show increased structural flexibility that favors accessibility to post-translational modification sites, and they also display a larger interaction surface area that allows them to interact with several ligands (*Uversky, 2015*). These traits are in fact characteristic of disordered proteins that are related to several cellular processes, such as cell signaling, transcription, and stress response (*Gsponer et al., 2008*; *Hincha & Thalhammer, 2012*). Moreover, experimental evidence and *in silico* analysis suggests that, in general, DHNs behave as IDPs in aqueous solution (*Graether & Boddington, 2014*).

IDPs have shorter protein half-lives than globular proteins, since they possess long intrinsically disordered regions that have been shown to be more susceptible to several degradation machineries, such as PEST degradation sequences (*Rechsteiner & Rogers, 1996*). The PEST sequences are one of the most common motifs for protein degradation. PEST regions are considered to be flexible, unstructured, and they contribute to protein disorder since they are enriched with amino acids such as proline, glutamic acid, serine, and threonine (*Rechsteiner & Rogers, 1996*).

There is evidence of a positive correlation between DHN protein accumulation and plant stress tolerance in *Arabidopsis thaliana*, *Solanum sogarandinum*, *Deschampsia antarctica* among others (*Hanin et al., 2011*; *Szabala, Fudali & Rorat, 2014*; *Olave-Concha et al., 2004*); however, DHN protein degradation has not yet been explored. We have previously reported that the cold-inducible *OpsDHN1* gene from Cactus pear encodes an IDP that is able to assemble into homodimers in both the cytoplasm and nucleus of tobacco cells (*Ochoa-Alfaro et al., 2012*; *Hernández-Sánchez et al., 2014*; *Hernández-Sánchez et al., 2015*). Herein, we characterize PEST sequences located in the central and C-terminal region of

the OpsDHN1 protein. For this aim, translational fusions derived from the *OpsDHN1* open reading frame and the β-glucuronidase (*GUS*) reporter gene were generated. These GUS::OpsDHN1 derived fusions were assessed by transient expression in *Nicotiana benthamiana* leaves, and also in stable *A. thaliana* transgenic lines. Histochemical and fluorometric analyses of GUS::OpsDHN1 fusions showed that the half-length, containing both central and C-terminal PEST sequences, is enough to reduce GUS protein stability through the 26S proteasome pathway. In order to demonstrate that these PEST sequences are functional, we designed a version of the OpsDHN1 that has the phosphorylatable residues of the PEST sequences mutated and showed that this reestablishes the GUS signal. Finally, we conducted an *in silico* analysis of PEST occurrence using 195 DHN orthologues, comprising all five DHN classes described so far, with the goal of identifying more potential PEST sequences.

## MATERIALS AND METHODS

### Plant material and growth conditions

To obtain tobacco plants, *Nicotiana benthamiana* seeds were sown on a mix of 50% vermiculite and 50% soil and incubated in a growth chamber with a photoperiod of 16 h light (120 μmol m$^{-2}$ s$^{-1}$) and 8 h darkness for 3–4 weeks.

Seeds of *Arabidopsis thaliana* ecotype Col-0 were used. First, seeds were sterilized with a 20% (v/v) chlorine solution for 5 min and then washed tree times with sterile distilled water. Next, seeds were germinated on Murashige and Skoog (MS) 0.5× plates, pH 5.7, containing 0.5% (w/v) sucrose, and 1% (w/v) agar (*Murashige & Skoog, 1962*). Seeds were stratified for 2 days at 4 °C in the dark, and then the plates were incubated at 22 ± 2 °C in a growth chamber with a 16 h light (120 μmol m$^{-2}$ s$^{-1}$) and 8 h darkness. After that plants were transferred to a mix of vermiculite and soil (1:1) during three weeks until its transformation.

### Vector generation

To generate the pMDC32-GUS control construct; first, the *GUS* open reading frame was amplified by PCR using the Phusion high-fidelity DNA polymerase (Invitrogen, Carlsbad, CA, USA). Subsequently, GUS amplicon was introduced into the pCR8/GW/TOPO entry vector (Invitrogen) and then it was sub-cloned into the pMDC32 plant expression vector (*Curtis & Grossniklaus, 2003*) by attL/attR recombination sites using Gateway LR Clonase II Enzyme Mix (Invitrogen).

Each pMDC32-GUS::OpsDHN1 (full-length, PEST-1, PEST-2, and PEST-1-2) derived construct was generated by the fusion of two PCR products, the first one containing a version of the *GUS* open reading frame without a stop codon, and the second one with a stop codon, either *OpsDHN1* open reading frame, *OpsDHN1* nucleotides 372-594 (PEST-1), nucleotides 561-747 (PEST-2) or nucleotides 372-747 (PEST-1-2). To fuse the *GUS* and *OpsDHN1* amplicons, the *Kpn* I recognition sequences were introduced in oligonucleotide sequence, and after PCR amplification these were digested with *Kpn* I enzyme (Invitrogen) generating cohesive ends and fused using T4 DNA ligase (Invitrogen). The ligated products

were cloned into the pCR8/GW/TOPO entry vector and then sub-cloned into the pMDC32 vector as mentioned before. Selected clones were sequenced using the M13 forward primer.

## Site-directed mutagenesis

The GUS::PEST-mut construct was synthesized *de novo* by GenScript (Piscataway, NJ, USA). Serine (S), threonine (T), and tyrosine (Y) codons in PEST sequence were replaced to alanine (A) as indicated: in the first positive PEST sequence, serine codons (TCG and TCT) were replaced by GCG and GCT, respectively, tyrosine codon (TAC) was replaced by GCA; in the second positive PEST sequence, both serine codons (TCA) were replaced by GCA, and threonine codon (ACT) was replaced by GCT; in the negative poor PEST sequence three threonine codons (TAC) were replaced by GCA. The substitutions were designed in order to produce the minor changes in codon sequence. To perform gateway cloning system, the attL/attR recombination were included flanking the GUS::PEST-mut version. The attL-GUS::PEST-mut-attR DNA fragment was cloned into pUC57 vector between *Hind* III and *Bam* HI enzyme restriction sites. Mutated construct was confirmed by sequencing and then it was sub-cloned into the pMDC32 vector (*Curtis & Grossniklaus, 2003*) using Gateway LR Clonase II Enzyme Mix (Invitrogen).

## Plant transformation

The tobacco leaves were infiltrated with *A. tumefaciens* GV3101 strain carrying the pMDC32 expression vectors (*Belda-Palazón et al., 2012*). The *A. tumefaciens* cells were grown until they reached an $OD_{600}$ of 1.0. Cells were collected and re-suspended in an equal volume of infiltration buffer (10 mM $MgCl_2$, 10 mM MES pH 5.6, 200 μM acetosyringone). Then cell suspensions were incubated in continuous shaking for 3 h at 28 °C. Each Agrobacterium strain containing the pMDC32 constructs was infiltrated in the abaxial side of tobacco leaves using a syringe without needle (*Belda-Palazón et al., 2012*). Transformed leaves of three plants were used to perform GUS histochemical staining and fluorometric assays after 3 days of infiltration. All experiments were repeated three times for each construct observing similar results.

The "floral dip" method was used (*Zhang et al., 2006*), the *A. tumefaciens* GV2260 strain containing the appropriate vector were employed to generate *A. thaliana* transgenic lines. The transformed plantlets were identified on 0.5× MS medium supplemented with 50 mg/mL hygromycin B. From this, seven different *A. thaliana* transgenic lines were obtained for all constructs. The L1 line was chosen for the pMDC32-GUS control vector, and L1 and L2 lines were chosen for the pMDC32-GUS::OpsDHN1 full-length, pMDC32-GUS::PEST-1, pMDC32GUS::PEST-2, and pMDC32-GUS::PEST-1-2 derived fusions. Twelve-days-old seedlings of T3 generation were used for all the experiments.

## GUS histochemical and fluorometric analyses

Before GUS analysis, the transient-transformed tobacco leaves were detached from the plant and cut in circles of one inch diameter, and whole *A. thaliana* 12-day-old transgenic seedlings were taken off from 0.5× MS plates using forceps. Samples were incubated in GUS staining buffer (0.5 mg/mL 5-bromo-4-chloro-3-indol-β-D-glucoronide in 100 mM sodium phosphate at pH 7.0) at 37 °C for 12 h (*Jefferson, Kavanagh & Bevan, 1987*). Finally

the chlorophyll was removed as described by Malamy and Benfey (*Malamy & Benfey, 1997*). The *A. thaliana* GUS staining images were acquired using a 10×/0.25 dry objective with a microscope (MOTIC BA-300) coupled to a 5.0 megapixel camera. Tobacco GUS stained samples were captured under a stereomicroscope (MOTIC SMZ-143) at 2× magnification. The basic photo program from the OS X system (Apple, CA, USA) was used to analyze all images. Three independent experiments were carried out obtaining similar results.

For the fluorometric assay, the plant tissues from agro-infiltrated *N. benthamiana* leaves and *A. thaliana* transgenic seedlings were harvested, frozen, and homogenized in 0.5 mL extraction buffer (50 mM sodium phosphate, pH 7.0, 10 mM 0.1% (w/v) dithiothreitol, 0.1% (w/v) Triton X-100, sodium lauroyl sarcosine, 10 mM EDTA). After centrifugation at 13,000 rpm for 10 min, GUS activity was assayed at 37 °C in the supernatant using 1 mM 4-methylumbelliferyl- β-D-glucuronide (4-MU) as substrate. The reaction was neutralized adding 0.2 mL of 200 mM sodium carbonate. Next, fluorescence was quantified using a TECAN GENios microplate fluorometer (Tecan, Shanghai, China) with a fixed excitation source (365 nm) and an emission filter (460 nm). Protein concentrations were determined by Bradford assay (*Bradford, 1976*).

## 26S proteasome inhibition

Inhibition of the 26S proteasome in *A. thaliana* seedlings was performed on MS liquid medium supplemented with 150 μM MG132 (Sigma-Aldrich, St. Louis, MO, USA) for 3 h. MG132 stock solution (5 mg/mL) was prepared with dimethylsulfoxide (DMSO). 0.5× MS liquid medium and 0.5× MS liquid medium containing DMSO were employed as controls. After treatments, the *A. thaliana* seedlings were processed to GUS histochemical and fluorometric analyses.

## RNA isolation and RT-PCR analysis

The RNA isolation of *A. thaliana* seedlings was carried out using Concert reagent (Invitrogen) according to manufacturer's instructions. The RNA samples were treated with DNase I (Invitrogen). One microgram of total RNA was used to synthesize the cDNA using the Super Script II enzyme (Invitrogen) according to manufacturer's recommendations.

For RT-PCR analysis, a 400 bp fragment of the *GUS* reporter was amplified and 154 bp of the *A. thaliana AtUBQ5* ubiquitin gene was used as the loading control. The oligonucleotides used for *GUS* amplification were as follows: 5-GUS 5'-atgttacgtcctgtagaaaccccaacc-3' (sense) and 3-GUS 5'-cacaaacggtgatacgtacact-3' (antisense). For *UBQ5* amplification, the oligonucleotides were used as follows: 5-UBQ5 5'-tcgacgcttcatctcgtcctc-3' (sense) and 3- UBQ5 5'-ggatctggaaaggttcagcg-3' (antisense). The semi-quantitative RT-PCR analysis was carried out in a 25 μL reaction containing 250 ng of cDNA template, 0.5 μL of each primer, 2.5 μL 10X buffer, 1.5 μL 50 mM MgCl$_2$, 0.5 μL 10 mM dNTP's, and 0.5 μL of recombinant Taq polymerase (Invitrogen).

RT-PCR amplification conditions of *GUS* reporter were: 5 min at 94 °C, 28 cycles of 30 s at 94 °C, 30 s at 60 °C, and 30 s at 72 °C, followed by 5 min at 72 °C. For the expression analysis of *UBQ5*, PCR conditions were the following: 5 min at 94 °C, 28 cycles of 30 s at 94 °C, 30 s at 60 °C, and 30 s at 72 °C, followed by 5 min at 72 °C.

## Identification of PEST regions in Dehydrins

The 195 DHN sequences were obtained from Phytozome (*Goodstein et al., 2012*) and GenBank (*Benson et al., 2013*) databases. Sequences were analyzed by using the ePEST-FIND program (emboss. bioinformatics.nl/cgi-bin/emboss/epestfind). The algorithm identifies stretches of 10 or more amino acids, enriched in proline (P), glutamic acid (E), aspartic acid (D), serine (S), and threonine (T), and flanked with the positively charged amino acids arginine (R), histidine (H), and lysine (K). This parameter is combined with hydropathy index to obtain a PEST score as expressed by the following equation: PEST score = 0.55 * DEPST-0.5 * hydrophobicity index (*Rechsteiner & Rogers, 1996*). Only those PEST sequences with positive scores were reported.

## Statistical analysis

Student's $t$-test analysis was carried out to determine statistically significant differences between tobacco cell expressing the pMDC32-GUS and pMDC32-GUS::OpsDHN1 constructs. One-way ANOVA and Tukey's multiple comparison post-test analyses were performed to evaluate statistical significance of GUS activity among plant cells expressing pMDC32-GUS construct, pMDC32-GUS::OpsDHN1 full-length, pMDC32-GUS::PEST-1, pMDC32-GUS::PEST-2, pMDC32-GUS::PEST-1-2, and pMDC32-GUS::PEST-mut derived fusions. Two-way ANOVA and Tukey's multiple comparison pos $t$-test were performed to analyze statistical significance among of GUS activity of pMDC32-GUS and pMDC32-GUS::OpsDHN1 derived versions under MS, MS+MG132, and MS+DMSO treatments. The GraphPad Prism version 5.0b (GraphPad Software, San Diego, California, USA) was used. All data represent the mean $\pm$ SEM ($n = 3$). Different letters on the bars represent means that are statistically different at $P < 0.05$.

## RESULTS

### Prediction of PEST sequences in OpsDHN1 protein

In order to identify potential PEST sequences in the OpsDHN1 protein, we used the ePEST-FIND algorithm (http://emboss.bioinformatics.nl/cgi-bin/emboss/epestfind). Based on the local enrichment of critical residues (hydrophilic and negative charged) and hydrophobicity index of PEST sequences, ePEST-FIND generates a value ranging from about −50 to +50. PEST scores greater than zero are considered significant (*Rechsteiner & Rogers, 1996*). This scrutiny disclosed two significant PEST sequences with positive scores of +3.8 and +8, respectively; they were localized at the central and C-terminal domains of OpsDHN1. The first one was located within residues 146 to 166 [HVEEVIYSEPSYPAPAPPPPH; PEST-1], between the first and second K-segments; the second PEST sequence was found from 237 to 248 residues [KDVECDQPPSST; PEST-2], after the third K-segment at the end of the polypeptide chain (Fig. 1).

### The OpsDHN1 full-length fused to GUS facilitates its proteolytic degradation

In order to determine whether the OpsDHN1 PEST sequences function as a proteolytic target, we constructed a translational fusion between the stable *GUS* reporter gene

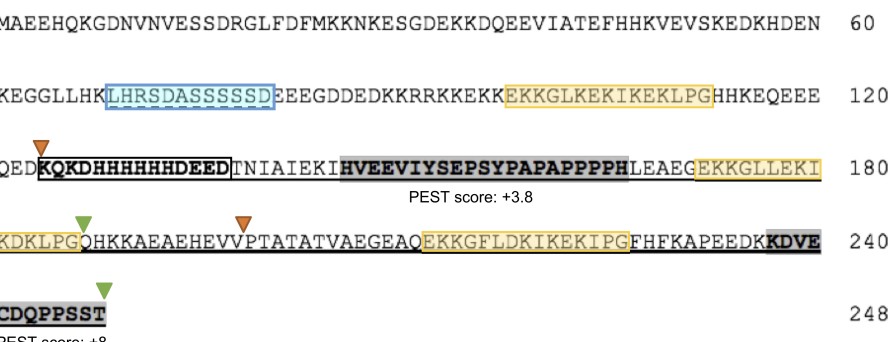

**Figure 1** **The *Opuntia streptacantha* OpsDHN1 amino acid sequence and schematic representation of its characteristic motifs and PEST sequences.** The OpsDHN1 conserved motifs are shown as follows: The S- and K-segments (blue and yellow boxes, respectively), histidine-rich motif (in bold and open box). Positive PEST sequences are in bold and inside gray boxes. The PEST scores are indicated below them. The half-length OpsDHN1 PEST containing region used to generate the GUS::PEST-1-2 construct is underlined. The orange and green triangles indicate the regions fused to GUS in the GUS::PEST-1 and GUS::PEST-2 construct, respectively.

and the *OpsDHN1* full-length nucleotide sequence (GUS::OpsDHN1) (Fig. 2). The GUS::OpsDHN1 fusion was expressed transiently in *Nicotiana benthamiana* leaves and in *Arabidopsis thaliana* transgenic lines. The *GUS* open reading frame was used as an expression control (Fig. 2A). Our histochemical data revealed that agro-infiltrated tobacco leaves and *A. thaliana* 12-day-old seedlings carrying the GUS control construct showed a strong GUS signal (Figs. 2B and 2D). However, plant cells harboring the GUS::OpsDHN1 construct displayed a lower GUS signal in contrast to the GUS control (Figs. 2B and 2D). The fluorometric analyses revealed a 70% decrease in GUS enzyme activity in plant cells that express the GUS::OpsDHN1 fusion relative to those that harbor the GUS control (Figs. 2C and 2E).

To ascertain if the low GUS signal observed in Arabidopsis plants harboring the GUS::OpsDHN1 construct was due to a post-translational regulation rather than a transcriptional control, we conducted RT-PCR assays in the *A. thaliana* GUS L1 control line and in two GUS::OpsDHN1 transgenic lines (L1 and L2) (Fig. S1). Our data showed similar *GUS* transcription levels among the analyzed *A. thaliana* transgenic lines, which suggest that the low GUS signal observed in those *A. thaliana* cells expressing the GUS::OpsDHN1 fusion was due to a post-translational regulation.

## The individual PEST-containing region of OpsDHN1 fused to GUS reporter lead to its degradation

In order to analyze each positive PEST sequence, we generated two GUS translational constructs: GUS::PEST-1 comprised of residues 124 to 198 of OpsDHN1, corresponding to the central PEST sequence; and GUS::PEST-2, comprised of residues 187 to 248, corresponding to the C-terminal PEST sequence. Each fusion was analyzed using the previously described tobacco and Arabidopsis expression systems (Fig. 3A). Our histochemical data revealed a GUS signal reduction on plant cells harboring both fusions

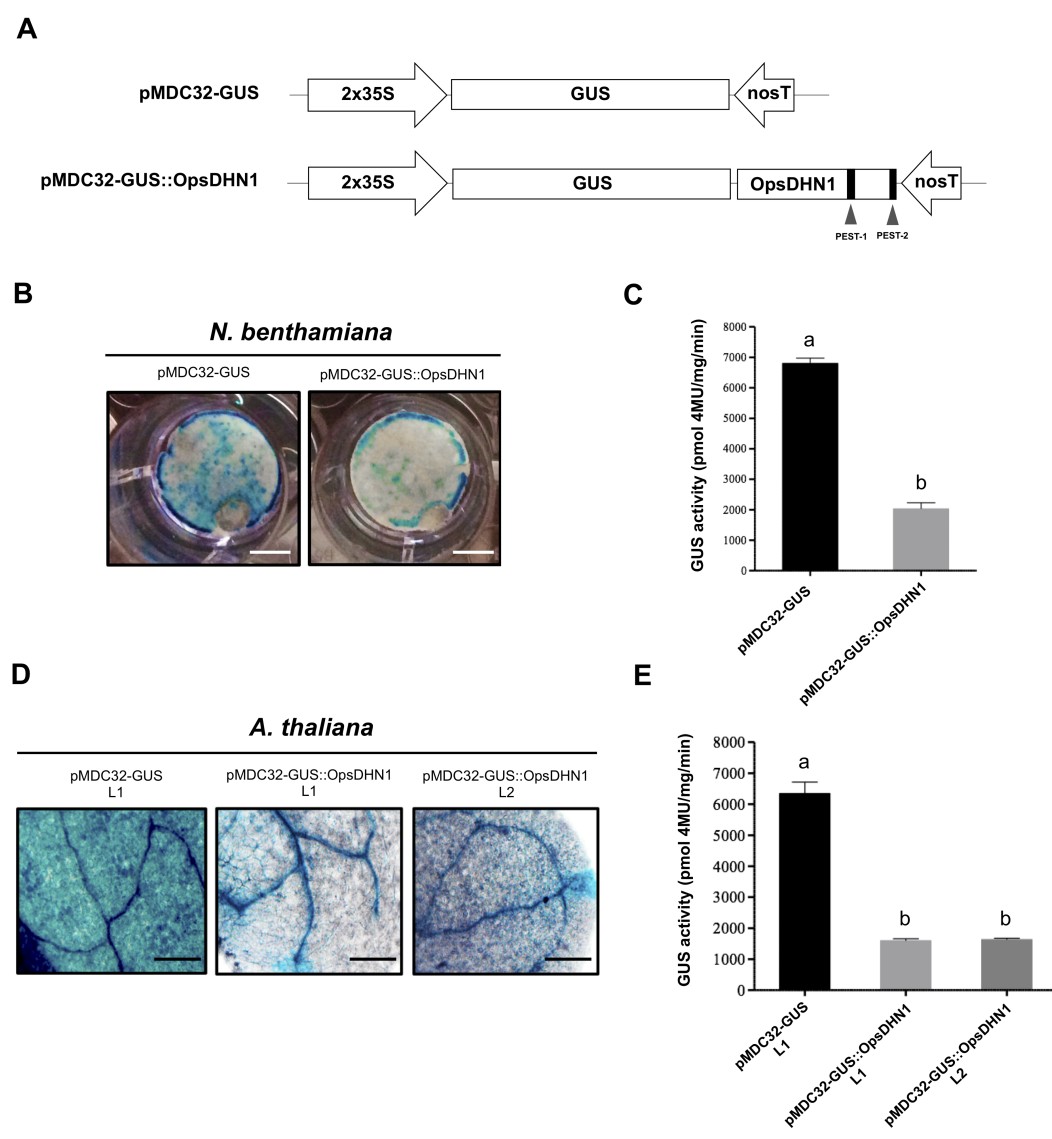

**Figure 2  The OpsDHN1 fused to GUS leads to its proteolytic degradation.** (A) Schematic representation of the pMDC32-GUS control and pMDC32-GUS::OpsDHN1 full-length constructs. The PEST sequences are marked in black boxes. (B–C) GUS histochemical and fluorometric assays in *N. benthamiana* leaves and (D–E) *A. thaliana* transgenic plants carrying the pMDC32-GUS control and pMDC32-GUS::OpsDHN1 full-length constructs. Representative images of GUS staining are shown in each column. The tobacco (B) and Arabidopsis (C) images were acquired at 2× and 10× magnification using a stereomicroscope and light microscope, respectively. The scale bar corresponds to 2,500 and 100 μm, respectively. GUS activity is reported as pmol 4MU/mg/min. Error bars represent the mean ± SE ($n = 3$). Letters indicate significant differences of GUS activity between plant cells expressing the pMDC32-GUS and pMDC32-GUS::OpsDHN1 constructs according to Student's *t*-test analysis ($P < 0.05$) (C), and one-way ANOVA and Tukey's multiple comparison post-test analyses ($P < 0.05$) (E). The experiments were repeated at least three times for each construct with similar results.

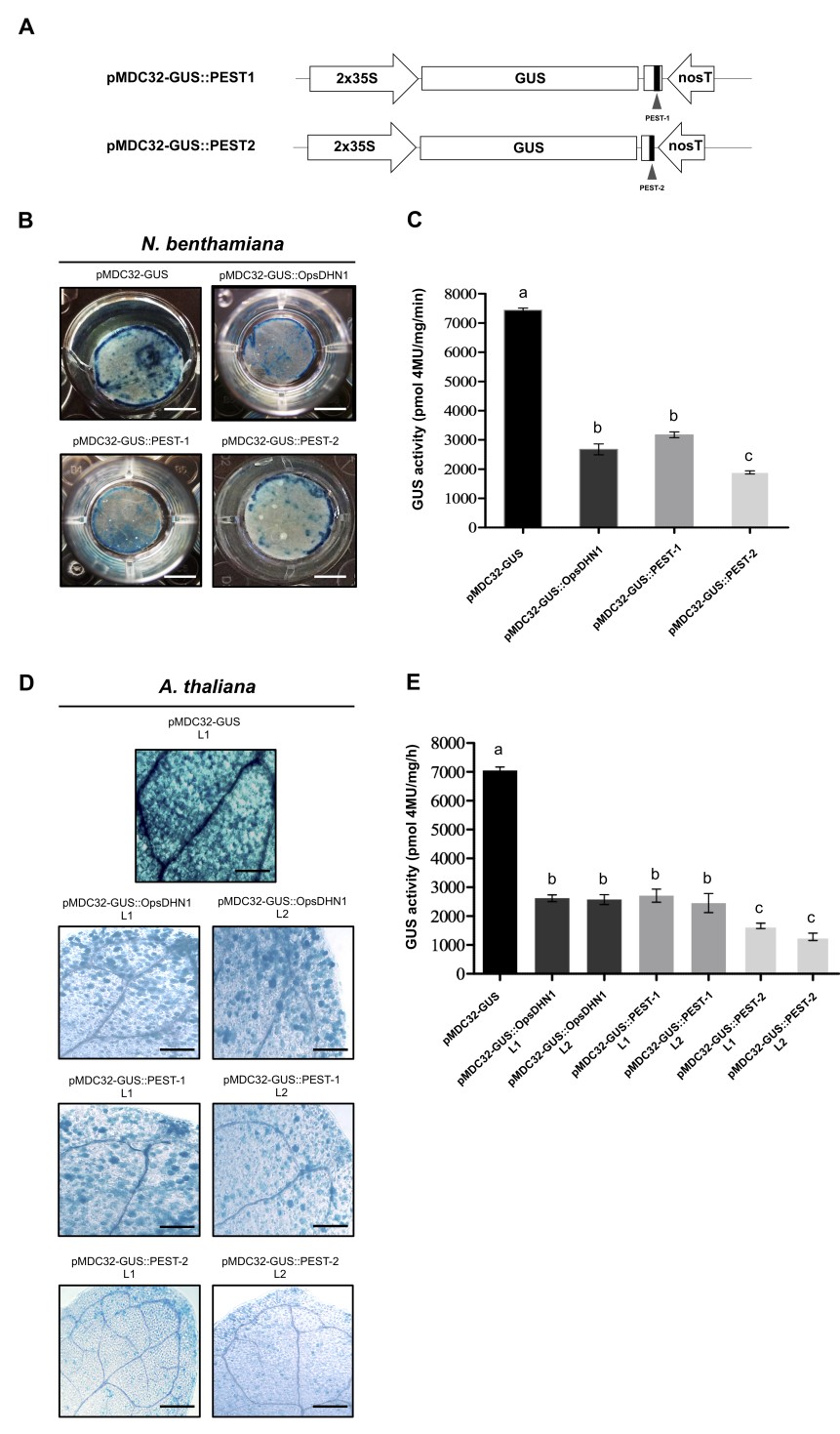

**Figure 3** **The OpsDHN1 PEST-containing regions fused to GUS leads to its proteolysis.** (A) Schematic representation of the pMDC32-GUS::PEST-1 and pMDC32-GUS::PEST-2 constructs. The PEST sequences are represented as black boxes. GUS histochemical and fluorometric assays in *N. benthamiana* leaves (B–C) and *A. thaliana* transgenic plants (D–E) (continued on next page…)

**Figure 3 (…continued)**
carrying the pMDC32-GUS, pMDC32-GUS::OpsDHN1, pMDC32-GUS::PEST-1, and pMDC32-GUS::PEST-2 constructs. Representative images of GUS staining are shown in each column. The tobacco (B) and Arabidopsis (C) images were acquired at 2× and 10× magnification using a stereomicroscope and light microscope, respectively. The scale bar corresponds to 2,500 and 100 μm, respectively. GUS activity is reported as pmol 4MU/mg/min. Error bars represent the mean ± SE ($n = 3$). Letters indicate differences in GUS activity among those plant cells expressing the pMDC32-GUS, pMDC32-GUS::OpsDHN1, pMDC32-GUS::PEST-1, and pMDC32-GUS::PEST-2 constructs according to one-way ANOVA analysis and Turkey's multiple comparison tests ($P < 0.05$). The experiments were repeated at least three times for each construct with similar results.

(GUS::PEST-1 and GUS::PEST-2), in comparison to the cells expressing only the GUS reporter. This decrement in GUS signal was relative to that observed in cells expressing the GUS::OpsDHN1 full-length sequence (Figs. 3B and 3D). The fluorometric assay revealed that the GUS activity was significantly diminished in plants cells that express the GUS::PEST-2 construct in comparison to those plants harboring the GUS::PEST-1 and GUS::OpsDHN1 full-version (Figs. 3C and 3E). RT-PCR assays showed that the *GUS* expression levels in the *A. thaliana* GUS::PEST-1 lines (L1 and L2) and GUS::PEST-2 lines (L1 and L2) were similar to the GUS L1 control line and GUS::OpsDHN1 lines (L1 and L2) (Fig. S1). These data show that presence of a single PEST is enough for protein degradation and reinforce our previous observation that protein degradation is due to the presence of the PEST regions.

## The OpsDHN1 region that contains PEST sequences enhances GUS degradation

We evaluated the effect of the two OpsDH1 PEST regions on the GUS stability. *GUS* nucleotide sequence was fused in frame with the last 375 bp from *OpsDHN1* gene that includes both PEST regions (GUS::PEST-1-2) (Fig. 4A). Both histochemical and fluorometric GUS assays were performed in tobacco and Arabidopsis cells that express the GUS::PEST-1-2 fusion. Our data showed that the GUS signal was not detected in any plant expression systems during the histochemical test (Figs. 4B and 4D). These results were in accordance to the fluorometric approach where the GUS activity was abated in comparison to the plant cell extract harboring the GUS control construct (Figs. 4C and 4E). We analyzed *GUS* expression in the *A. thaliana* GUS::PEST-1-2 lines (L1 and L2) through RT-PCR assays. Our results indicate that the GUS::PEST-1-2 (L1 and L2) lines and *A. thaliana* control line showed similar *GUS* expression levels (Fig. S1). These data reveal that the fusion of a region that contains two OpsDHN1 PEST sequences leads to complete GUS degradation.

## The 26S proteasome pathway mediates the proteolytic degradation of GUS::OpsDHN1 generated fusions

To test whether the 26S proteasome pathway is implicated in the degradation of GUS::OpsDHN1 fusions, we performed 26S proteasome inhibition assays using the *A. thaliana* GUS control line and those expressing the GUS::OpsDHN1 full-version (L1 and L2 lines) and GUS::PEST-1-2 (L1 and L2 lines). Twelve-day-old plants were incubated in MS liquid medium (Fig. 5A), and MS liquid medium supplemented with 150 μM

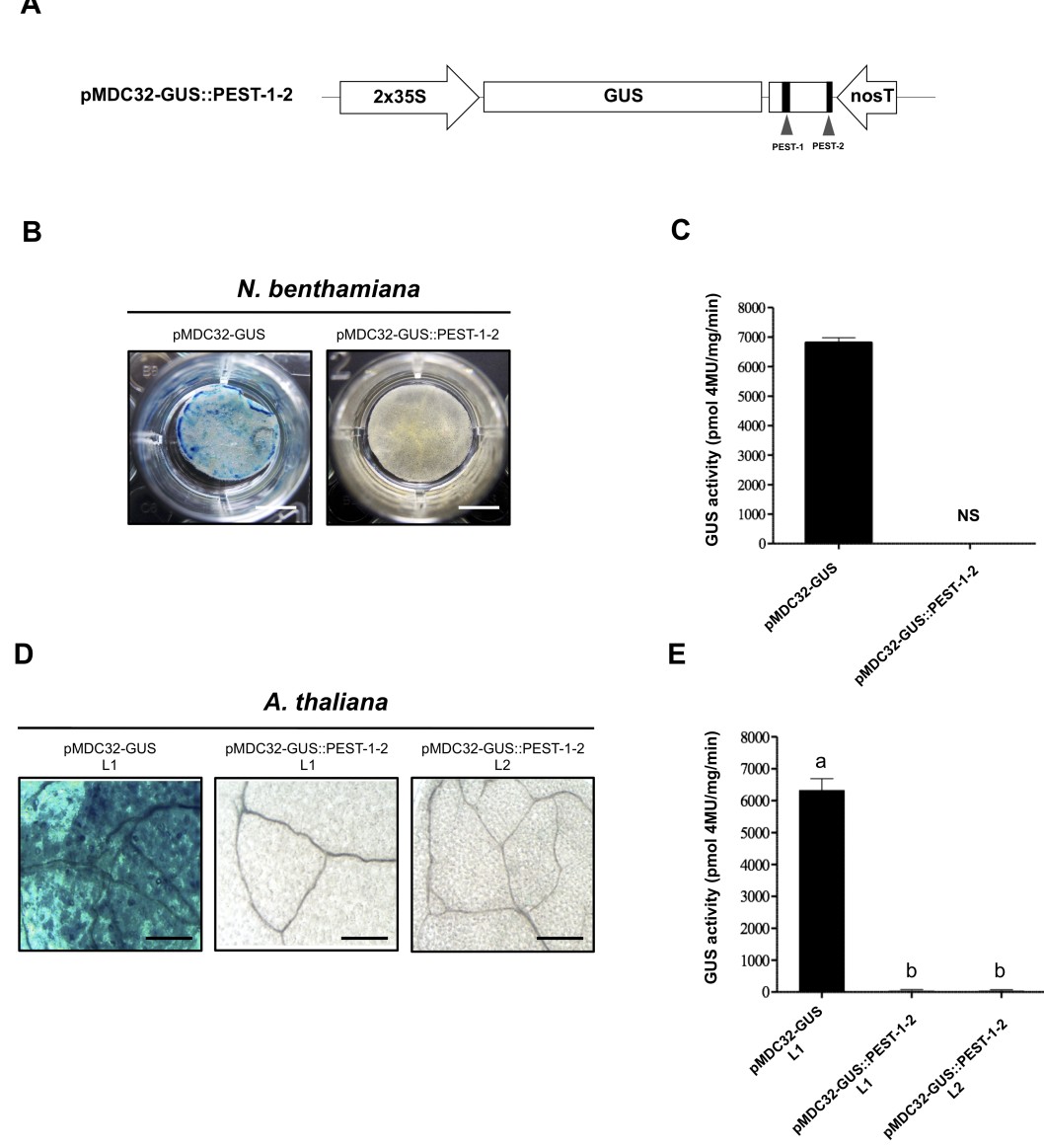

**Figure 4  The OpsDHN1 C-terminal PEST-containing region leads to complete GUS degradation.** (A) Schematic representation of the pMDC32-GUS::PEST-1-2 construct. The OpsDHN1 PEST sequences are marked in black boxes. GUS histochemical and fluorometric assays in *N. benthamiana* leaves (B–C) and *A. thaliana* transgenic plants (D–E) carrying the pMDC32-GUS control and pMDC32-GUS::PEST-1-2 constructs. Representative images of GUS staining are shown in each column. The tobacco (B) and Arabidopsis (C) images were acquired at $2\times$ and $102\times$ magnification using a stereomicroscope and light microscope, respectively. The scale bar corresponds to 2,500 and 100 $\mu$m, respectively. GUS activity is reported as pmol 4MU/mg/min. Error bars represent the mean $\pm$ SE ($n = 3$). Letters indicate significant differences of GUS activity between plant cells expressing the pMDC32-GUS and pMDC32-GUS::PEST-1-2 constructs according to the one-way ANOVA analysis and Tukey's multiple comparison tests ($P < 0.05$). NS: not signal. The experiments were repeated at least three times for each construct with similar results.

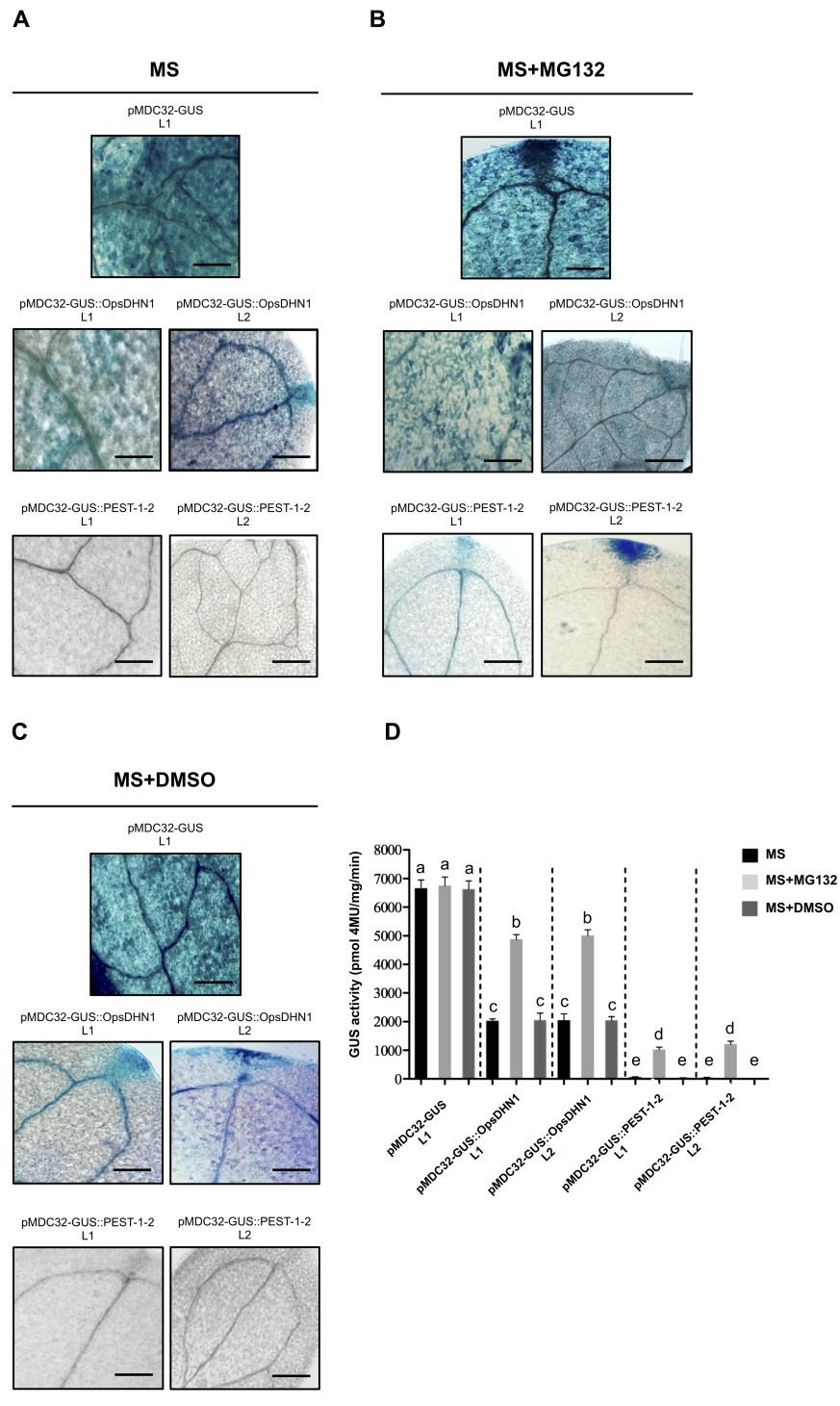

**Figure 5  The MG132 treatment increases GUS signal in *A. thaliana* the pMDC32-GUS::OpsDHN1 full-length and pMDC32-GUS::PEST-1-2 lines.** The *A. thaliana* 12-day-old lines were incubated in (A) MS liquid medium, (B) MS liquid medium supplemented with 150 μM MG132, and (C) MS liquid medium with DMSO, for 3 h. Representative images of GUS staining (continued on next page...)

**Figure 5 (...continued)**
are shown in each column. The Arabidopsis images were acquired at 10× magnification using a light microscope. The scale bar corresponds to 100 μm. (D) GUS activity of *A. thaliana* lines incubated in MS, MS+MG132, and MS+DMSO treatments was measured by fluorometric quantification of 4-MU. Bars indicate the mean ± SEM ($n = 3$) of GUS activity expressed as pmol 4 MU/mg/h. Statistical significance was determined by two-way ANOVA and Tukey's multiple comparison tests ($P < 0.05$). Letters indicate the differences of GUS activity among those plant cells expressing the pMDC32-GUS, pMDC32-GUS::OpsDHN1, and pMDC32-GUS::PEST-1-2 constructs treated with MS, MS+MG132, and MS+DMSO. The experiments were repeated at least three times with similar results.

MG132 proteasome inhibitor (Fig. 5B); as MG132 is soluble in DMSO, MS liquid medium containing this solvent was included as a control treatment (Fig. 5C). The histochemical assays showed that MG132 treatments considerably increased the GUS signal in both *A. thaliana* lines harboring the GUS::OpsDHN1 and GUS::PEST-1-2 constructs (Fig. 5B) compared with treatments without inhibitor (MS and MS+DMSO control treatments) (Fig. 5D). In particular, the GUS::OpsDHN1 and GUS::PEST-1-2 *A. thaliana* lines showed a significant increase in GUS activity, of around 40%, in presence of MG132 compared to plantlets treated with MS and MS+DMSO (Fig. 5D). There were no significant differences in GUS signal when the GUS L1 control line was incubated in MS or MS with MG132 (Fig. 5D). These results reveal that the 26S proteasome pathway is involved on degradation of GUS::OpsDHN1.

## Abolishment of GUS degradation by point mutations in PEST sequences

Since the proteasome-mediated protein degradation is often activated by phosphorylation of serine (S), threonine (T), and tyrosine (Y) residues present in PEST sequences (*Rechsteiner & Rogers, 1996*); we performed the point mutation analysis of these residues on the OpsDHN1 PEST sequences. It is worth mentioning that we identified a third putative PEST sequence in OpsDHN1 protein; however, this PEST sequence [195-HEVVPTATATVAEGEAQEK-213, Fig. 1] has a negative score (−0.61), so it is considered a poor sequence in the ePEST-FIND algorithm. We constructed an OpsDHN1 mutated version (GUS::PEST-mut), which includes the following substitutions by alanine (A): for the PEST1 sequence Y−152−A, S−153−A, S−156−A and Y−157−A; for the PEST2 sequence S−246−A, S−247−A ; T−248−A; and for the poor PEST sequence T−200−A, T−202−A , and T−204−A (Fig. 6A). A fusion between *GUS* and the last 375 bp from *OpsDHN1* mutated version (GUS::PEST-mut) was carried out by *de novo* gene synthesis.

We examined the GUS control, GUS::OpsDHN1 full-version, GUS::PEST-1-2, and GUS::PEST-mut constructs, which were transiently expressed using the previously described tobacco system (Figs. 6B and 6C). Our histochemical and fluorometric analyses revealed a significant recovery of GUS signal in those tobacco cells transformed with the GUS::PEST-mut construct in contrast to the cells harboring the GUS::OpsDHN1 full-length and GUS::PEST-1-2 versions (Figs. 6B and 6C). In the same way, the *A. thaliana* stable expression system showed increased GUS activity in plants expressing the GUS::PEST-mut, even more than the full GUS::OpsDHN1 version (Figs. 6D and 6E).

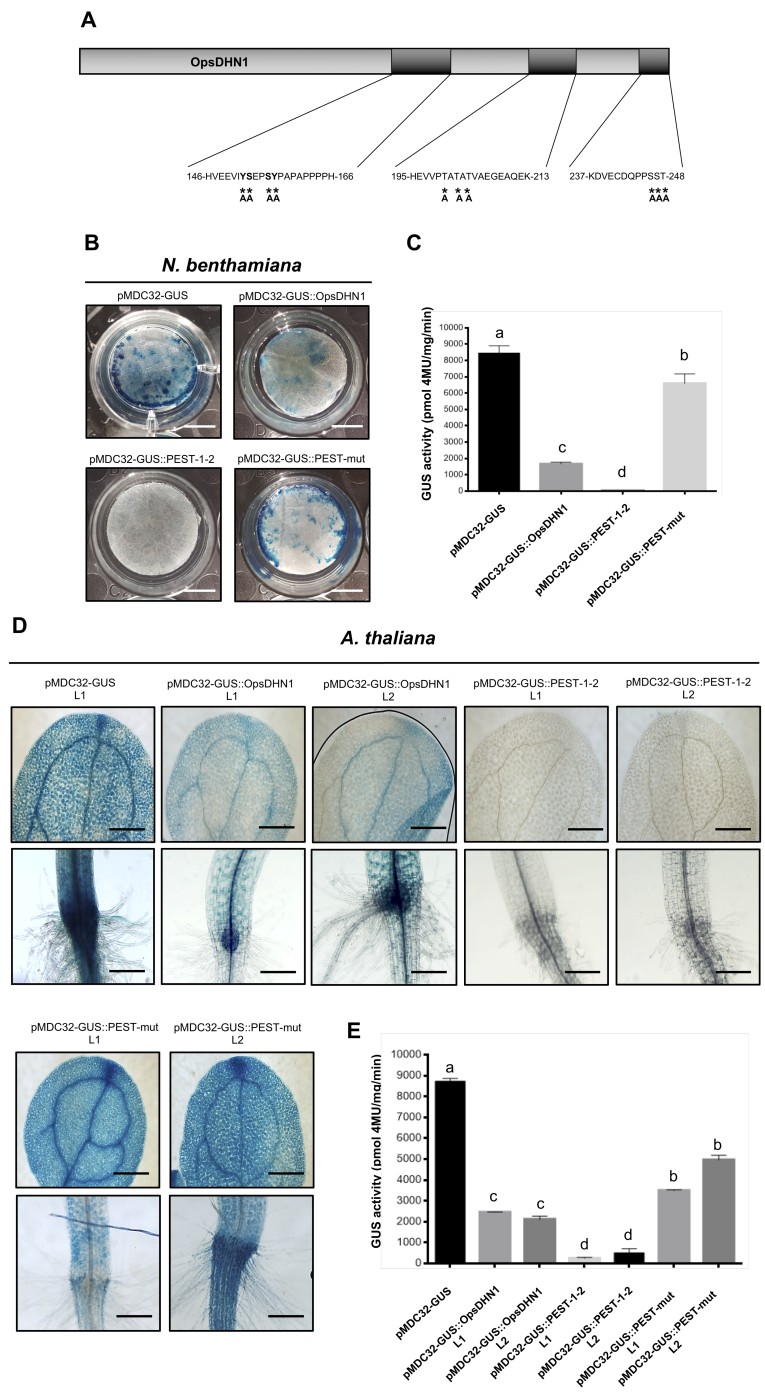

**Figure 6** **PEST sequence mutations enhance GUS signal in plant cells harboring the pMDC32-GUS::PEST-mut version.** Schematic representation of PEST-OpsDHN1 site-directed mutagenesis. Serine (S)-153, -156, -246, -247; threonine (T)-200, -202, -204, -248; and tyrosine (Y)-152, -157 spanning in the PEST sequences were substituted with alanine (A) residues. Histochemical and fluorometric GUS assays of pMDC32-GUS control, pMDC32-GUS::OpsDHN1, pMDC32-GUS::PEST-1-2, and pMDC32-GUS::PEST-mut constructs in *N. benthamiana* leaves (B–C) and *A. thaliana* transgenic plants (D–E). The scale bar corresponds to 2,500 and 100 $\mu$m, respectively. GUS activity is reported as pmol 4MU/mg/min. Error bars represent the mean $\pm$ SE ($n = 3$). Letters indicate significant differences of GUS activity according to the one-way ANOVA analysis and Tukey's multiple comparison tests ($P < 0.05$). The experiments were repeated at least three times for each construct obtaining similar results.

**Table 1** Distribution of potential PEST sequences in subgroups of DHN orthologs.

| DHN subgroup | Total number of proteins | Number of PEST-containing proteins | % of PEST-containing proteins |
|---|---|---|---|
| $SK_n$ | 48 | 36 | 75 |
| $Y_nSK_n$ | 50 | 12 | 24 |
| $K_n$ | 37 | 12 | 32.4 |
| $Y_nK_n$ | 29 | 8 | 27.5 |
| $K_nS$ | 31 | 0 | 0 |
| Total | 195 | 68 | 34.8 |

The expression of the GUS::PEST-mut fusion in *A. thaliana* lines (L1 and L2) was analyzed as previously described by RT-PCR assays. Our results indicate that the GUS::PEST-mut expression levels were similar to the GUS expression in the *A. thaliana* control line (Fig. S1). These data indicate that phosphorylation residues present in OpsDHN1 PEST sequences are key targets for its 26S proteasome proteolytic pathway.

### PEST sequence occurrence in DHNs

Finally, we investigated the occurrence of PEST sequences in DHN orthologues. We analyzed a total of 195 DHN protein sequences, based on their architectures: 48 $SK_n$, 50 $Y_n SK_n$, 37 $K_n$, 29 $Y_nK_n$, and 31 $K_nS$, using the ePEST-FIND algorithm. In general, 68 of the total DHNs examined contain at least one PEST sequence with a positive score (Table 1). Particularly, the $SK_n$ group was the most common PEST subclass, comprised of 36 proteins with positive PEST scores (Table 1). On the other hand, the remaining DHN classes exhibited a lower number of PEST-containing proteins. For example, the $Y_n SK_n$ subclass contained 12 proteins with a potential PEST sequence, $K_n$ had 12, and $Y_nK_n$ had 8; remarkably, in the case of the $K_nS$ type, no potential PEST-containing proteins were found (Table 1; Table S1).

### DISCUSION

Dehydrins (DHNs) play fundamental roles on plant stress response and adaptation (*Hanin et al., 2011*). Accumulation of DHNs have been used as molecular marker of plant abiotic stress; comparative studies among plant cultivars and varieties with marked differences in stress tolerance revealed a positive correlation between *DHN* gene expression and DHN protein accumulation, and also plant stress tolerance (*Hanin et al., 2011*). *In vitro* experiments have shown that some DHNs bind to lipid vesicles containing acidic phospholipids, bind to metals and have radical-scavenging properties, suggesting their ability to protect membranes against lipid peroxidation and also display cryoprotective activities toward freezing-sensitive enzymes (*Graether & Boddington, 2014*).

The biosynthesis of stress response proteins (e.g., chaperones, COR/LEA/DHNs, ROS scavenging enzymes), represents a high-energy requirement for the plant cell (*Kosová et al., 2011*). During plant stress recovery, degradation of these stress proteins is essential for maintaining a correct cellular function, re-establishment of growth and development, and

maintaining an adequate amino acid pool (*Araújo et al., 2011*; *Correa Marrero, Van Dijk & De Ridder, 2017*). Recovery of seedlings from stress seems to depend on their ability to degrade such stress proteins. Therefore, understanding LEA protein degradation is important. *Chourey, Ramani & Apte (2003)* observed a growth arrest of young seedlings of rice (Cv *Bura Rata*) after recovery from salt stress treated for 10 days in comparison to seedlings treated for a shorter period, which resumed their normal growth after NaCl treatment. The authors suggested that this growth arrest seems to depend on the inability of the plant to degrade LEA proteins. The marked accumulation of LEA proteins during stress and their necessary degradation after stress supports the idea that these stress-response proteins must be finely regulated to restart plant growth and development after stress.

The increases in DHN transcript and protein levels are closely related to the plant's ability to tolerate abiotic stress (*Kosová, Vítámvás & Prášil, 2014*). It is known that both *DHN* transcript and protein levels decline after stress has been overcome, as shown with DHNs from Siberian spruce (*Picea obovata*), barley (*Hordeum vulgare*), and birch (*Betula pubescens* Ehrh) (*Kjellsen et al., 2013*; *Du et al., 2011*; *Welling et al., 2004*). Despite the importance of DHNs, their post-translational regulation has not yet been explored. Here, we reported the proteolytic regulatory motifs in the Cactus pear OpsDHN1 protein. The *in silico* analyses revealed two potential PEST tags, with scores of +3.8 and +8, encoded in the central and C-terminal regions of OpsDHN1, respectively. The histochemical stain performed in Arabidopsis transgenic lines expressing the GUS::OpsDHN1 full-length fusion revealed a strong decrease in the GUS signal compared to those lines harboring the GUS control. In plants, the PEST degradation sequences have been functionally characterized in transcription factors. *Yamaguchi et al. (2010)* reported that the *A. thaliana* transcriptional repressor VNI2 has a C-terminal PEST sequence (score: +4.9), and the fusion of the full-length VNI2 protein to GUS led to specific proteolytic degradation in Arabidopsis transgenic plants. Likewise, *Sakuma et al. (2006)* reported the specific proteolytic degradation of GFP through the fusion with *A. thaliana* full-length DREB2A transcription factor (GFP::DREB2A), which contains an N-terminal PEST sequence (score: +9). Also, it has been reported that the CBF/DREBs family regulates the expression of several stress response genes, including *DHN* genes (*Vazquez-Hernandez et al., 2017*).

In order to characterize each of the OpsDHN1 PEST sequences, we generated GUS::PEST1 and GUS::PEST2 constructs. Each of these fusions showed a similar decrease in GUS activity in *N. benthamiana* leaves and *A. thaliana* transgenic lines, as observed in GUS::OpsDHN1 full-length fusion. However, the GUS signal was completely abated when we fused the half of OpsDHN1 that includes the two PEST sequences (GUS::PEST-1-2 construct). These results confirm that both PEST sequences are functional *in planta* and necessary for the degradation of OpsDHN1.

It is well known that the ubiquitin-proteasome system is essential to enable plants to change their proteome in order to respond to environmental stresses in an effective and efficient way (*Stone, 2014*). Interestingly, *Gumilevskaya & Azarkovich (2010)* suggested the involvement of the ubiquitin-proteasome system in the degradation of a 50 kDa DHN from horse chestnut. In addition, authors demonstrate that this 50 kDa DHN cross-reacted

with an antibody against ubiquitin, inferring that this DHN is ubiquitinated. The PEST-dependent proteolysis via 26 proteasome has been reported in the cases of CDKB2 N-terminal region (which contains a PEST sequence with a score of +9.7) from Arabidopsis and also in the C-terminal domain of *Zea mays* ZmSPMS1 spermine synthase (which contains a PEST sequence with a score of +3.6) using the MG132 proteasome inhibitor to block the degradation process (*Adachi, Uchimiya & Umeda, 2006*; *Maruri-López et al., 2014*). Likewise, after incubation with the MG132 inhibitor, we observed a GUS signal recovery in the *A. thaliana* transgenic plants carrying the GUS::OpsDHN1 full-length, and half-length OpsDHN1 protein that comprises PEST sequences. These data show that the OpsDHN1 is in fact degraded via the ubiquitin-proteasome system.

A decrease in the content of LEA and even DHN proteins after drought stress is not uncommon. In Siberian spruce, protein levels of the 53 kDa DHN reach the highest levels during winter and decrease rapidly after freezing stress (*Kjellsen et al., 2013*). In addition, *Vaseva et al. (2011)* compared the gene and protein profile in two drought-resistant pastures *Trifolium pratense* and *T. repens*; the authors observed that under dehydration stress two DHN transcripts and their respective proteins are markedly expressed and accumulated, and after recovery both signals returned to the control levels (*Vaseva et al., 2011*).

DHNs have evolved to maintain high flexibility and avoid aggregation/denaturation (*Hincha & Thalhammer, 2012*). It has been reported that disordered regions of proteins correlate with post-translational modification sites, such as phosphorylation and ubiquitination/proteasomal degradation (*Tompa, 2002*; *Kurotani & Sakurai, 2015*). PEST regions have been predicted to serve as rapid degradation signals (*Sandhu & Dash, 2006*). Herein, we found that the fusion of OpsDHN1 full-length to GUS is more stable in comparison to the GUS::PEST-1-2 construct (harboring half of the OpsDHN1 protein). It is tempting to think that this stability is due to the full-version of the OpsDHN1 have the ability to form homodimers. Previously, we reported that OpsDHN1 dimerizes in yeast and tobacco cells (*Hernández-Sánchez et al., 2014*; *Hernández-Sánchez et al., 2015*); also, the deletion of regions containing K-segments in OpsDHN1 reduces its dimerization in yeast cells (*Hernández-Sánchez et al., 2014*). Interestingly, it has been reported that PEST regions participate in protein-protein interactions avoiding its proteolytic degradation, such is the case of the interaction between the PEST-containing *A. thaliana* S-adenosyl-methionine decarboxylase 1 (AtSAMDC1) and the viral suppressor C2 protein encoded by Beet Severe Curly Top Virus (BSCTV). The dimerization of BSCTV C2/AtSAMDC1 proteins could regulate the AtSAMDC1 degradation to provide a hypomethylated environment that promotes viral accumulation (*Zhang et al., 2011*). Likewise, the mammalian ornithine decarboxylase (ODC), which contains PEST sequences in its C-terminal domain, is stable in a homodimeric conformation (*Zhang et al., 2004*); and in maize spermine synthase ZmSPMS1 homodimer formation involves its C-terminal region, which contains a functional PEST sequence (*Maruri-López et al., 2015*). Based on these data, we propose that the OpsDHN1 dimerization could also be regulating its stability.

PEST sequences are frequently conditional proteolytic tags, and PEST-carrying proteins are not degraded until they are labeled. Particularly, phosphorylation has been described to promote degradation process on proteins containing PEST sequences (*Rechsteiner &*

*Rogers, 1996*; *Penrose et al., 2004*); these phosphorylation events occur on serine, threonine, and tyrosine residues followed by ubiquitination and rapid degradation by the proteasome pathway (*Rechsteiner & Rogers, 1996*; *Penrose et al., 2004*). In order to analyze the role of phosphorylatable residues, we carried out the point mutation analysis of these residues on the three OpsDHN1 PEST sequences (GUS::PEST-mut). By replacement of the corresponding serines, tyrosines, and threonines with alanines, we obtain more evidence supporting that the OpsDHN1 PEST sequences are functional, since this version mutated in the three PEST construct was able to recover GUS signal in both plant expression systems, suggesting that the phosphorylation is key for OpsDHN1 degradation.

It is clear that protein turnover is fundamental for cellular survive during and after stress response. Based on the *in silico* identification of possible PEST sequences in the five-DHN classes, we can suggest that the case of OpsDHN1 is not unique, and that DHNs of the $SK_n$ family could have functional PEST targets. Nevertheless, the 26S proteasome pathway degradation is conserved and is the most used mechanism for protein degradation in eukaryotes (*Karsies, Hohn & Leclerc, 2001*); this machinery is fed by a variety of degradation tags (*Kats et al., 2018*). DHNs are highly evolved proteins that have been adapted their protein sequences to respond to particular stress response and/or plant development stages, so the presence of other degradation signals should not be discarded in other groups of DHNs. Detailed *in silico* and *in vivo* DHNs degradation sequences analyses are needed to help to dissect how these enigmatic proteins are regulated.

## CONCLUSION

This study is the first report on PEST degradation regions in the intrinsically disordered DHN family. We show that the OpsDHN1 PEST sequences are functional tags for degradation of the target fusion protein, suggesting a functional role in the OpsDHN1 turnover *in planta*. Our results provide evidence supporting that OpsDHN1 degradation depends on the phosphorylation of serine, threonine, and tyrosine residues present in the PEST sequences. The presence of PEST sequences in DHNs opens a new scenario in the study of the post-translational regulation of these proteins.

## ACKNOWLEDGEMENTS

The authors gratefully acknowledge Dr. Steffen Graether and Karamjeet Singh for their critical reading and grammatical review.

### Funding

This work was supported by CONACYT (Proyectos de Desarrollo Científico para atender Problemas Nacionales, 2015-01-414). The funders had no role in study design, data collection and analysis, decision to publish, or preparation of the manuscript.

## Grant Disclosures

The following grant information was disclosed by the authors:
Proyectos de Desarrollo Científico para atender Problemas Nacionales: 2015-01-414.

## Competing Interests

The authors declare there are no competing interests.

## Author Contributions

- Adriana L. Salazar-Retana and Israel Maruri-López performed the experiments, prepared figures and/or tables, authored or reviewed drafts of the paper, approved the final draft.
- Itzell E. Hernández-Sánchez performed the experiments, analyzed the data, prepared figures and/or tables, authored or reviewed drafts of the paper, approved the final draft.
- Alicia Becerra-Flora performed the experiments, contributed reagents/materials/analysis tools, prepared figures and/or tables, authored or reviewed drafts of the paper, approved the final draft.
- María de la Luz Guerrero-González performed the experiments, contributed reagents/materials/analysis tools, prepared figures and/or tables, authored or reviewed drafts of the paper, approved the final draft.
- Juan Francisco Jiménez-Bremont conceived and designed the experiments, performed the experiments, analyzed the data, contributed reagents/materials/analysis tools, prepared figures and/or tables, authored or reviewed drafts of the paper, approved the final draft.

## Data Availability

The raw data is available as a Supplemental File.

## Supplemental Information

Supplemental information for this article can be found online at http://dx.doi.org/10.7717/peerj.6810#supplemental-information.

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
