# Peer review of "PEST sequences from a cactus dehydrin regulate its proteolytic degradation"

_PeerJ, doi:10.7717/peerj.6810_

## Round 0.1 · original submission · Minor Revisions

In general the manuscript was well written and had significant findings which will be of value to the research community. The reviewers have outlined their concerns which should be easily addressed. I will classify this manuscript as requiring “minor” revisions; however, as pointed out by one reviewer there appears to be an opportunity to expand upon these findings and to provide a broader view of the significance regarding the role of some/all dehydrins. In general, there may be some fine-tuning with some of the sentence structures. A few that I saw are listed here:

line 77: “motives” to “motifs”.
line 255: “construct is due” to “construct was due”.
line 368: “to plant’s ability” to “to the plant’s ability”.
line 424: “OpsDHN1 has the ability” to “OpsDHN1 to have the ability”.
line 431: “Beet curly top virus” to “Beet Curly Top Virus”.
line 434: “the mammal” to “the mammalian”.
line 449: “three PEST was” to “three PEST constructs was”.
line 458: “suggesting its could regulate” to “suggesting it could regulate”.
line 461: “open new scenarios” to “open a new scenario”.

Reviewer 1 ·

Basic reporting

The manuscript is clearly and well written overall. However, the writing of “Materials and Methods” session need improve due to the use of non-standard English. For example, “scattered”, “a proportional vermiculite and soil mix”, “after they were…’’ in the paragraph (line 103-105), and “shifted into” in multiple places, were not properly worded.

Other comments related to writing:
Line 54, “suggest” should be “suggests”
Line 56, what is the superscript of “segment6”?
Line 56 and 77, “motives” should be “motifs”
Line 62, “present” may be changed to “contain”?
Line 355 , “represent” should be “represents”
Line 361, remove “,” after et al.
Line 450, rephrase “ transient tobacco leaves”
Line 457, “protein fusion” should be “fusion protein”
Line 458, “its” should be “they”
Line 636, “plant cell” should be “ plant cells”

Table 1, orthologous should be “orthologs”. The title may be changed to “ Distribution of potential PEST sequences in subgroups of DHN orthologs”

Legend of Figure 5, “in A. thaliana the pMDC32-GUS…..” should be “ in the A. thaliana pMDC-GUS…..”

Experimental design

Meet standards. No other comments.

Validity of the findings

no comment

Additional comments

Figure 6D and 6E, in staining (6D), the “pMDC-GUS::PEST-mut L1 and L2 seem to have high GUS activities than the control pMDC32-GUS L1; however, in the fluorometric assay (6E), the result indicates the opposite, please discuss in the manuscript.

Reviewer 2 ·

Basic reporting

Pass.

Experimental design

Pass.

Validity of the findings

Pass.

Additional comments

The methods are very clearly explained, with credible results shown in the figures and supplemental information..The findings significantly add to the body of knowledge on dehydrins, as noted by the authors addressing an aspect of dehydrins - degradation during stress-recovery - that has received little attention so far. So, PEST sequences seem to be part of the picture. It leaves the reader wondering if/why it is not always part of the picture for every dehydrin, or at least always part of the picture for some specific subset of dehydrins. Perhaps this could be addressed somewhat more in the Discussion.

As a routine practice, each DNA construct should be checked by sequencing to make sure that the intended junctions have been made. For some constructs this validation was stated. For most it was not. The authors should insert statements to confirm that all constructs were validated by sequencing.

---

## Round 0.2 · accepted · Accept

All of the suggestions were sufficiently attended to and the manuscript appears ready to be forwarded for publication. Congratulations, please consider this manuscript accepted.